# Ultrahigh strength and shear-assisted separation of sliding nanocontacts studied in situ

Takaaki Sato [1✉], Zachary B. Milne[2], Masahiro Nomura [3], Naruo Sasaki [4], Robert W. Carpick[1] & Hiroyuki Fujita[3,5]

The behavior of materials in sliding contact is challenging to determine since the interface is normally hidden from view. Using a custom microfabricated device, we conduct in situ, ultrahigh vacuum transmission electron microscope measurements of crystalline silver nanocontacts under combined tension and shear, permitting simultaneous observation of contact forces and contact width. While silver classically exhibits substantial sliding-induced plastic junction growth, the nanocontacts exhibit only limited plastic deformation despite high applied stresses. This difference arises from the nanocontacts' high strength, as we find the von Mises stresses at yield points approach the ideal strength of silver. We attribute this to the nanocontacts' nearly defect-free nature and small size. The contacts also separate unstably, with pull-off forces well below classical predictions for rupture under pure tension. This strongly indicates that shearing reduces nanoscale pull-off forces, predicted theoretically at the continuum level, but not directly observed before.

[1] University of Pennsylvania, Department of Mechanical Engineering and Applied Mechanics, Philadelphia, PA, USA. [2] Sandia National Laboratories, Nanostructure Physics, Albuquerque, NM, USA. [3] University of Tokyo, Institute of Industrial Science, Tokyo, JP, Japan. [4] The University of Electro-Communications, Department of Engineering Science, Tokyo, JP, Japan. [5] Tokyo city university, Graduate school of integrative science and engineering electrical and electronic engineering, Tokyo, JP, Japan. ✉email: takaakis@seas.upenn.edu

Understanding and predicting the behavior of materials in sliding contact is highly challenging, particularly because of the breadth of physical phenomena that can occur, all at a confined interface normally hidden from direct view[1,2]. For metal-metal interfaces, the presence of strong adhesion (once contaminants and/or oxides are removed, which can readily occur in sliding contacts) combined with applied normal and/or shear loads can often lead to ductile flow, strain hardening, and cold welding[3,4]. Unraveling and understanding these phenomena can help develop better models for describing and predicting the behavior of practical engineering interfaces, which consist of multitudes of such asperity contacts due to the typical roughness of surfaces. Metal-metal contacts are important as they often occur in machinery and engines, and also important in a wide range of fields including transportation, power conversion, manufacturing, medical device implants, and micro- and nanoelectromechanical system (MEMS/NEMS).

Extensive studies have used atomic force microscopy (AFM) and related techniques to explore single-asperity nanotribology, with the advantage that single asperity behavior can be measured with nano-scale force and displacement resolution and provide a well-defined contact geometry[5,6]. While the contact is not directly observed in standard AFM systems, intermittent characterization of the tip size and shape, and indirect measurements of physical quantities related to contact size such as contact stiffness, contact conductance, and adhesion provide valuable information on the contact geometry[5,6].

While the behavior of nanoscale metallic junctions subjected to normal loading has been studied by AFM or AFM-based methods, e.g., refs.[7–10], fewer studies have probed metal-metal nanocontacts in frictional sliding. In one example, a probe was slid across a metal surface such as Au and Cu with a load of less than 1 nN[11]. Stick-slip instabilities in the frictional force with atomic regularity were observed, but with low dissipation when low loads were used. This was attributed to facile shear of (111) planes in a metallic neck that the authors hypothesize was formed between the Cu(100) sample and the Si tip. At higher loads, irregular stick-slip and higher friction occurred, similar to previous measurements on Cu(100) surfaces[12].

Directly observing the shape and size of a contact as it slides could elucidate how sliding is accommodated, including the extent of elastic deformation and the degree of agreement at the nanoscale of continuum contact models such as the Johnson-Kendall-Roberts (JKR)[13] or the Derjaguin-Müller-Toporov (DMT)[14] models, whose validity at this scale is an ongoing research topic[15,16]. Observing inelastic behavior, such as dislocation activity and plastic flow, material transfer, and cold welding would enable determination of the deformation regime of the materials for robust models of asperity-level contact. This is substantially more challenging than studies of pure compressive or tensile loading, since inelastic deformation modes in combined loading are less well understood, particularly at the small scales. Such observations would also permit determining the nanoscale origins of important phenomena known or inferred in macroscale contacts such as junction growth[17,18], debris/third body formation[19,20], tribofilm/transfer-film formation[21–24], grain refinement[25], and subsurface crack formation[26]. Knowledge of the contact geometry is essential for estimating the stresses, which is required to determine interfacial shear strengths as well as values for yield and fracture stresses.

Recent developments in in situ instrumentation have enabled observation of contacting and sliding asperities in real time using transmission electron microscopy (TEM)[3,27–32]. Although standard TEM imaging cannot normally capture the full 3-D contact geometry, the 2-D image is sufficient to, for example, characterize a Pt-Pt contact accurately enough to correlate it with contact resistance measurements and confirm the presence and character of adsorbates[33], and to determine that a Au-Au contact diffused together in a liquid-like manner when far below the melting point of Au, and further, exhibited transfer of material from one surface to another after contact separation[30]. However, to date, TEM-based systems could not measure frictional and normal forces at the same time while also directly observing the actual contact area at the atomic scale.

We previously presented results and analysis of in situ nanoscale Ag-Ag single-asperity experiments, where two isolated Ag protrusions on opposing surfaces are brought into contact via lateral sliding using a custom-designed apparatus housing a nanoelectromechanical systems (NEMS) device, where only the lateral force was measured[29]. In this work, a NEMS device is developed to measure normal and shear forces simultaneously, and contact formation, progressive shear, and separation are all observed while monitoring the normal and shear forces. This reveals multiple phenomena at play in Ag-Ag contacts never before observed including spontaneous junction formation, unstable slip, the yield stresses reaching near-ideal strength behavior, and the separation force falling well below predictions used for pure tensile loading.

## Results

**Measurement approach**. Figure 1 shows a schematic of the apparatus, and a sequence from one of six in situ TEM tests conducted of a single nano-asperity nanocontact, extracted from real-time video (examples from five other experiments are shown in Supplementary Figs. 1–15). The bottom asperity was fixed laterally (it is connected to the cantilever which deflects vertically, in response to contact forces) and the upper asperity was actuated from left to right, as indicated in Fig. 1. The intersection of the plane of contact with the viewing plane gives the contact width. The normal force and shear force reported here are defined as the forces perpendicular and parallel to the plane of contact respectively, at each point of measurement (since the asperity is curved, the orientation of the plane of contact changes as sliding proceeds).

**Sliding experiments**. The asperities are initially not in contact (Fig. 1, c, i). The load became negative upon of contact (Fig. 1, c ii) due to the van der Waals attraction between the two asperities pulling them into adhesive contact. This is seen as a rapid jump-to-contact instability, consistent with snap-in behavior commonly seen in AFM experiments, and is a result of the attractive force gradient exceeding the normal stiffness of the system, which is primarily determined by the stiffness of the load cantilever. In this experiment, the load increased slightly (became less negative) when the upper asperity climbed over the lower asperity (Fig. 1, c, ii, iii), and then decreased (became more negative) when it traveled down the right-side slope of the lower asperity (Fig. 1, c, iii-iv). The friction force increased not only as the upper asperity climbed up the bottom asperity (Fig. 1, c, ii, iii) but as the upper asperity traveled down the other side of the bottom asperity (Fig. 1, c, iv, v). The surfaces then separate (Fig. 1, c, vi).

The shape of each asperity prior to contact formation was compared with that after contact separation. Fig. 2a shows the shape of lower asperity before contact, Fig. 2b the shape after separation, and Fig. 2c a comparison between the two. The apparent volume of the lower asperity increased slightly. To confirm this shape change, the same area on the same pair of asperities were traced to see whether the shape change also occurred with an increase of the actual contact area. The shape of the lower asperity exhibited deformation, so either material was added or plastic deformation of the lower asperity occurred (or both). As the volume change is so

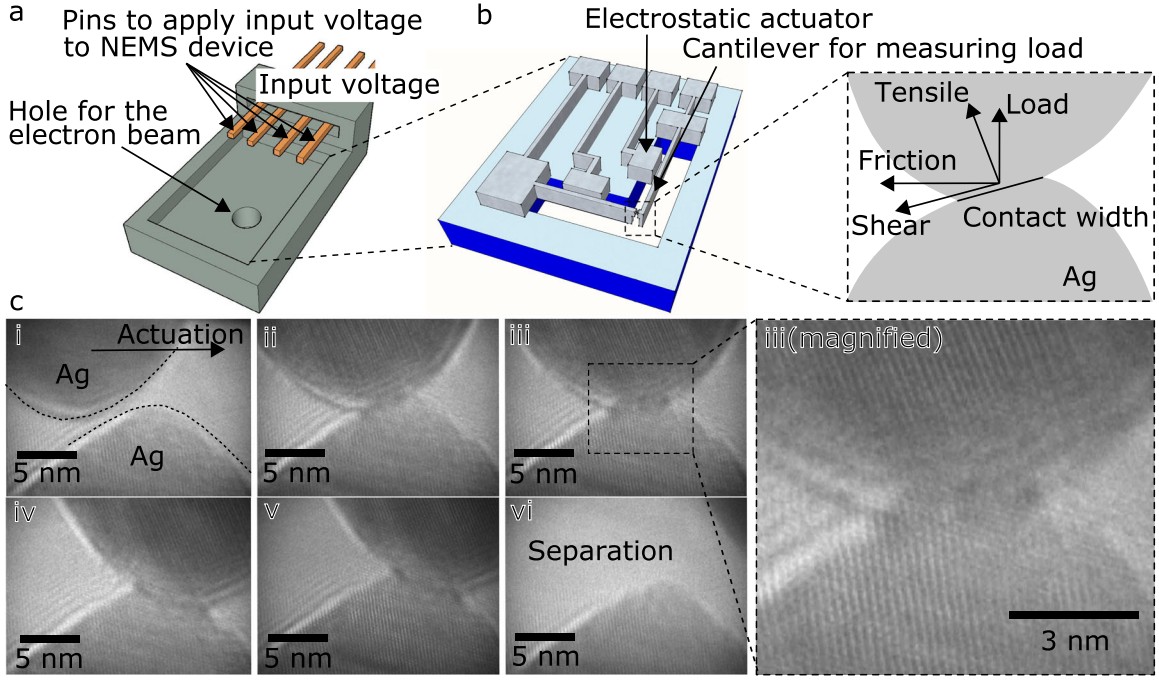

**Fig. 1 A custom-designed in situ apparatus enables observation of nanoscale single asperity friction[68,69]. a** Schematic of the stainless steel frame that holds the NEMS device, which is mounted at the sample location of a TEM holder. Four gold wires, used to drive the two electrostatic NEMS actuators, are shown along with a hole for passage of the TEM beam. **b** Schematic of the silicon-based NEMS device, showing two orthogonally-oriented cantilevers to measure friction and normal forces, and electrostatic actuators to move asperities in lateral and vertical directions. The inset shows the TEM view of the contact. The upper asperity is the one connected to the cantilever for measuring the friction force. **c** Example of a single asperity sliding experiment observed by TEM. i, The upper asperity is actuated in the lateral direction. Initially, the asperities are not in contact. ii, The lower asperity has been pulled into tensile contact with the upper asperity due to attractive forces. iii, iv, v, The upper asperity slides laterally across lower asperity. vi, The junction separates. Videos of this experiment are available (See Supplementary Movie).

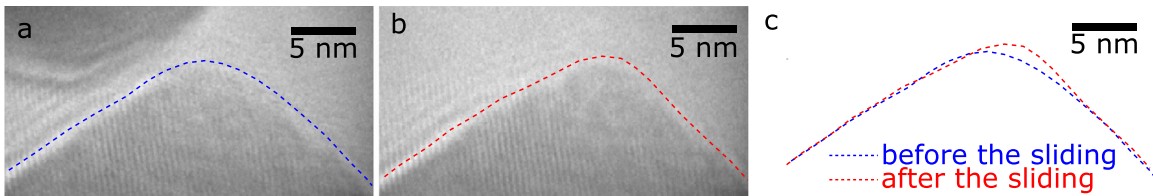

**Fig. 2 TEM images demonstrate that only limited nanoscale plastic deformation occurred due to contact, sliding, and separation.** For the experiment depicted in Fig. 1, the shape before the contact **a** was compared with the shape after the separation **b**, **c** depicts the difference between **a**, **b**. The lines shown are traced manually while magnifying the image. Despite the high stresses, the plastic displacements are less than 1.0 nm in size.

small, approximately 20 nm³ assuming locally axisymmetric symmetry at each measured height of the asperity, it is not possible to tell which. Regardless, the change in volume is small, demonstrating the precision with which plastic deformation can be resolved in this apparatus. Sliding experiments using the same procedure were performed for six trials. In all experiments, at most a nanoscale-level amount of deformation and/or transfer was observed (see Supplementary Figs. 1, 4, 7, 10, and 13).

**Extracting Stresses from Force and Contact Width**. Figure 3 shows the forces and mean stresses at the actual contact interface (i.e. each force divided by contact area, assuming axisymmetric contact) as a function of the sliding distance. Results for the five other experiments are shown in Supplementary Figs. 3, 6, 9, 12, and 15. By recording the forces and contact size concurrently, we are able to calculate the mean normal (tensile in these experiments) and shear stress, measured by dividing the normal or lateral force respectively by the simultaneously obtained contact

area, determined by assuming the contact is axisymmetric. We also calculate the von Mises stress as a function of sliding distance. We use the mean stress values at the contact interface and assume plane stress; this is not as accurate as a full stress analysis would reveal, but comparisons with contact mechanics models shows that this approach provides reasonable estimates, particularly considering our experimental uncertainty. This is plotted in Fig. 3d. The von Mises stress is valuable as a measure of the tendency for the onset of plastic flow via the empirical von Mises yield criterion[34]. While it is not particularly useful at stresses beyond the onset of plasticity, it is still directly related to the strain energy density. Thus, we briefly provide a discussion of the evolution of von Mises stress in all experiments. The von Mises stress for the plane stress loading condition used here is given by:

$$S_{vM} = \sqrt{\sigma^2 + 3\tau^2} \qquad (1)$$

where the normal stress $\sigma$, and the shear stress $\tau$ are based on the normal and shear directions with respect to the interface at each

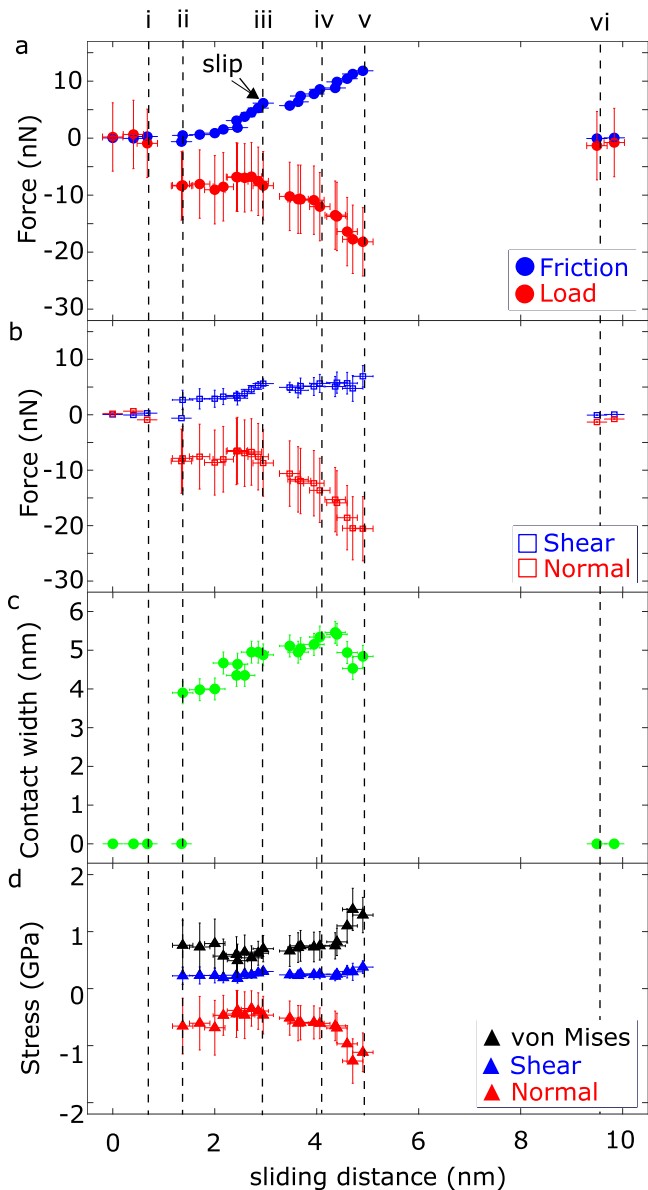

**Fig. 3 Forces (from the NEMS device), contact width (from TEM images), and resulting calculated stresses as a function of sliding distance.**
**a** Friction and load forces during an asperity friction measurement. The friction force acts parallel to the direction of the actuation, and the load force acts perpendicular to the friction force, as shown in Fig. 1. **b** In contrast, the shear force acts parallel to the plane of contact, whose orientation changes during sliding. Similarly, the normal force acts perpendicular to the plane of contact, i.e., perpendicular to the shear force (Fig. 1c, iv). **c** the contact width, measured as the shortest width of the junction. **d** The tensile normal stress, shear stress, and von Mises stress, derived from the values of the normal force, the shear force, and the contact width as shown in Eqs. (1)–(4). The indices i–vi corresponds to the panels in Fig. 1. A video of this experiment is available (See Supplementary Movie). The error bars representing the uncertainty of each experimental value arose from the resolution of the TEM and NEMS actuator, and the calculations are performed as described in Supplementary Discussion 2.

point (see Fig. 1c, iv), calculated as

$$\sigma = \frac{F_{normal}}{A} \qquad (2)$$

$$\tau = \frac{F_{shear}}{A} \qquad (3)$$

and the contact area

$$A = \pi \left(\frac{w}{2}\right)^2, \qquad (4)$$

where $ww$ is the contact width.

Figure 3d shows that the shear stress is relatively constant throughout the contact event. Because the contact is initially under tensile loading, the tensile normal stress reduces in magnitude as the upper asperity climbs over the lower asperity; this leads to a decrease in the von Mises stress as well. Then, both the tensile normal and the von Mises stress grow substantially in magnitude as the upper asperity descends down the lower asperity. This behavior was seen in three of the six experiments. In the other three, the tensile normal and von Mises stress show monotonic increases with sliding distance. The non-monotonic dependence is attributed to a combination of greater initial tensile normal stress due to an earlier pull-in instability and a more gradual increase of shear stress with sliding distance. The latter factor may be due to the specific arrangement of atoms at the contact interface and the precise misalignment of the two single crystal asperities.

## Discussion

In macroscopic contacts subjected to compressive and tangential loading (even without sliding, i.e., remaining in the static friction regime), ductile materials typically exhibit junction growth (substantial growth in the real contact area, sometimes of over 10 times)[33], due to plasticity induced by the applied shear stress; it can be readily shown from Eq. (1) that an increase in shear stress must lead to an increase in contact area for the von Mises stress to remain constant[34]. For example, Bowden and Rowe reported a 3-fold increase in the adhesion force between macroscopic silver asperities in vacuum after tangential loading, attributed to junction growth[35]. In stark contrast, junction growth in the present experiments is either not observed, or is very limited. Note that junction growth will still occur even though the normal stresses are tensile, not compressive, in these experiments, since the normal stress shows up quadratically in Eq. (1). Fig. 2c for example shows a small relative amount of plastic deformation, and similar or even smaller amounts are seen for the five other experiments as illustrated in the Supplementary Figs. 2, 5, 8, 11 and 14. The plastic deformation that is seen is consistently in the form of slight plastic necking near the top of the asperity and oriented in the direction of sliding. This indicates consistently that the level of ductility in these nanoscale junctions is substantially less than for macroscopic contacts, and correspondingly, that the yield strength is larger.

In all six experiments, a small instability, in the form of a jump in the sliding distance and a corresponding jump in friction and load forces, is seen at or shortly after point iii (Fig. 3, Supplementary Figs 3, 6, 9, 12, 15), i.e., the point where the tips of the two asperities align, i.e., when the asperity interaction transitions from climbing up to climbing down. Further similar instabilities are seen at other points, e.g., between points iv and v in Supplementary Fig. 9. Such jumps are similar to yielding events seen in uniaxial tensile tests that are due to plastic events such as dislocation nucleation, dislocation motion, or other more complex events involving (for example, stacking fault-based structures that are found in twinned Ag nanostructures[36]) and indicate unstable but limited plastic flow. Furthermore, Ag-Ag asperities of 3 nm radii (the same approximate size to those here) were subjected to contact and shear using molecular dynamics

recently[37]; they observed very similar instabilities which were directly a result of plastic deformation, supporting our contention that these instabilities are due to yield of the Ag. We note that the instabilities induced at the transition point between climbing up and climbing down may be due to hysteretic unloading behavior, where shrinkage of the contact area as the tensile stress magnitude increases (i.e., the unloading occurring when climbing down) is hindered by interfacial adhesion. In other words, during unloading, tensile elastic strain builds up, then is quickly released in a small strain burst as the contact suddenly shrinks; both elastic and plastic strain release may occur at this point.

Supplementary Table 1 lists the sliding displacement values and the corresponding stresses at each instability that could be readily identified by an observable, finite slip event. These plastic events occurred at von Mises stresses ranging from 0.70 to 1.78 GPa (mean value ± standard deviation of 1.29 ± 0.39), which corresponds to an effective shear stress ($1/\sqrt{3}$ times the von Mises stress) ranging from 0.40 to 1.03 GPa (mean value 0.74 ± 0.23 GPa). Considering only the initial yield event for each of the six experiments, the von Mises stresses range from 0.70 to 1.70 GPa (mean value 1.10 ± 0.39 GPa), which corresponds to an effective shear stress ranging from 0.40 to 0.98 GPa (mean value 0.63 ± 0.23 GPa). These initial yield stress values, plotted in Fig. 4 are large: these values are 18–22 times silver's typical bulk yield strength in pure tension of 50–60 MPa (in pure tension, the tensile stress and von Mises stress are identical)[36,38]. We compare these to the ideal strength of a crystal in the absence of defects such as dislocations and free surfaces can be roughly estimated to be $G/30$ (where $G = 27.8$ GPa for silver), i.e., 0.93 GPa[39], corresponding to a von Mises stress of 1.61 GPa (Fig. 4, blue line). Here, we obtain von Mises stresses at initial yield that reach 43–106% of this estimated ideal strength of silver. More precisely, the theoretical shear strength of pure Ag with no defects (including dislocations or surfaces) and at 0 K was calculated from density functional theory (DFT) by Ogata et al. to be 1.65 GPa, based on the {111}/<112> slip system[40], corresponding to a von Mises stress of 2.86 GPa (Fig. 4, red line). We observe von Mises stresses that reach 24–59% of this theoretical value, despite having proximal free surfaces and being measured at 300 K which can have the effect of reducing the yield stress from the theoretical value[41].

Although, as mentioned above, these von Mises stress values are approximate, they show that the asperities exhibit strength values under combined loading approaching a significant fraction of the ideal strength. This is surprising given that the asperities were formed after contact and separation via plastic flow and fracture, which one might expect would lead to the nucleation and propagation of a significant number of dislocations. The high strength observed may be in part due to the fact that plasticity in Ag tends to nucleate in the form of stacking faults (partial dislocation pairs) with a very low stacking fault energy[42] and thus a correspondingly large zone size, making them difficult to nucleate in small volumes.

We compare these to experimental measurements of yield of Ag at the nanoscale (Fig. 4). Experimental measurements of the yield of pure Ag nanowires (NW's) in uniaxial tension produced yield strength values of comparable magnitudes to ours. These nanowires are formed with a pentatwinned geometry, but have a low population of defects (other than the five longitudinal twin boundaries). For example, Zhu et al. reported a yield strength of such Ag NW's inside a scanning electron microscope (SEM), with the smallest silver nanowires (34–38 nm diameter) having a yield strength of approximately 2.5 GPa for loading along the <110> direction[43]. This approaches the theoretical strength Ogata et al. of 2.86 GPa[40,43]. Vlassov et al[44]. performed tensile tests on Ag NW's with diameters from tens to hundreds of nm and obtained

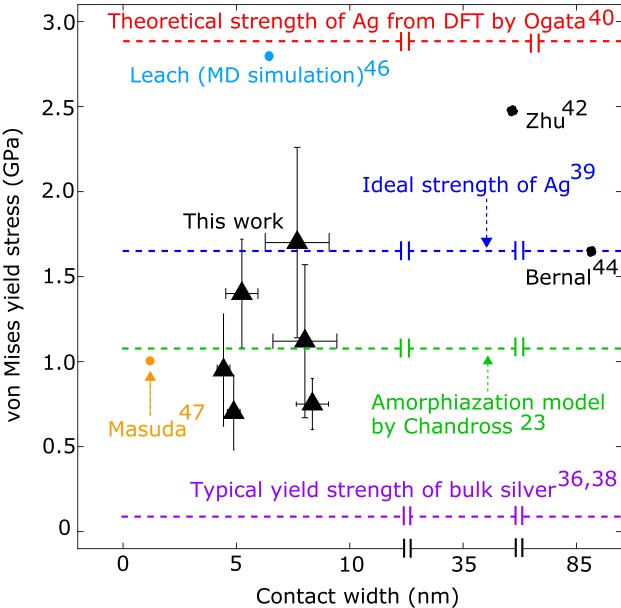

**Fig. 4 The experimental values of the von Mises stress at slip instabilities far exceed those of bulk silver.** They reached 24–59% of silver's theoretical strength according to DFT calculations[40], rivaling values observed for single crystal or pentatwinned nanowires in tension. Triangles: values of the von Mises stress measured at observed yield points. Dashed lines and circles show literature values discussed in the text. The error bars representing the uncertainty of each experimental value arose from the resolution of the TEM and NEMS actuator, and the calculations are performed as described in Supplementary Discussion 2.

a yield strengths ranging from 1–10 GPa, with a median reported value of 4.8 GPa. Bernal et al. performed tensile tests of Ag NW's 30 to 120 nm in diameter, observing yield stress values of approximately 1.6 GPa[45]. Similar measurements by Filleter et al. of Ag NW's 40 to 120 nm in diameter revealed yield stress values of 2–7 GPa[46]. These TEM results are particularly illuminating as they demonstrate explicitly the near perfection of the NW (such as the absence of dislocations), and that initial yield is correlated with particular plastic events, specifically, surface nucleation of stacking fault decahedrons. The results also showed that initial yield tended to occur at a particular strain value; the corresponding stress value was size-dependent, becoming larger at smaller scales due to a concomitant increase in the elastic modulus with size. MD simulations in those papers provided further support for the findings of high strength being associated with low defect populations at small scales. Leach et al[47]. conducted MD simulations of defect-free silver NW's with varying cross-sectional geometries, finding yield stress varied from 2.5 to 3.5 GPa both for penta-twinned NW's and for untwinned (single crystal) NW's. Masuda and Kizuka[48] performed TEM experiments of Ag-Ag nanoasperity contacts in tensile loading, whose contact widths were controlled down to the scale of one to two atoms. The found yield strengths of 0.5–0.6 GPa, and a tensile normal stress at final fracture of approximately 1 GPa.

These literature results are all plotted in Fig. 4. These comparisons illustrate that our observed von Mises stress at yield under combined loading of 0.70 to 1.78 GPa approach or even exceed yield stress values from experiments and simulations in pure tension of crystalline Ag nanostructures with low defect densities. However, in all but the last example above, our experimental geometry differs substantially as it involves two

single crystal asperities brought into contact, where the two crystals may be misaligned, as opposed to the loading of a single, long nanowire. More significantly, the experiments reported here are the first to explore in situ the combined role of tensile, normal, and shear stresses in the yielding of silver at the nanoscale; few such studies, where both shear and normal forces are resolved, exist for any materials.

The plastic yielding events mentioned above and indicated in Fig. 3 and the related plots in the Supplementary Figs produce lateral slip distances of 0.34–1.58 nm, with an average value of 0.76 nm. The Burgers vector for Ag is 0.2889 nm[49,50]. Thus, we observe slip distances that are of the order of one to three Burgers vectors in size. While this is consistent with the possibility that plastic slip within either or both asperities is the mechanism by which plastic flow is accommodated during these yielding events, we could not in general resolve individual dislocations in the TEM during sliding. Thus, it is also possible the concurrent slip of atomic planes within the asperity occurred, or slip occurred at the interface formed between the two asperities.

We now consider the contact of the two asperities in the context of how polycrystalline metals deform since, essentially, the contact is a bicrystal interface, with proximal surfaces. While very different compared to an infinite bicrystal or to a polycrystalline sample where there are no free surfaces in the vicinity of the interface, we may use this analogy by considering that there may be a transition in behavior as a function of asperity size. Namely, considering how the mechanics changes as asperities get smaller, there should initially be an effective Hall-Petch regime where preexisting dislocations become increasingly sparse, leading to a strengthening of the two asperities (dislocation-mediated plasticity). At smaller scales, an effective inverse Hall-Petch regime may occur, where the two-asperity system's strength is reduced. While there are multiple possible mechanisms invoked to explain the inverse Hall-Petch effect in polycrystals, we consider here the mechanism of grain boundary sliding.

Chandross et al. have shown that, at small scales, for potentially arbitrary metallic systems, grain boundary sliding can lead to amorphization of the material at and emanating from the interface[23]. In the Ag deposition process of our experiments, the upper and lower asperities form at a distance from each other and their growth is not epitaxially related, which means the crystal orientations of the asperities are not deterministically linked. The videos and Fig. 1 iii (magnified) clearly show that the apexes are misoriented and also suggest that, during sliding, the contact and some distance within each grain may be amorphous. The contact region may thus resemble a low-angle grain boundary, with an amorphous barrier due to the low energy barrier of the self-mixing of silver. We caution that, without proper exit wave reconstruction, interpretations of bright spots in the TEM image as atoms or interatomic voids can be misleading.

That all yield stresses are close to but often somewhat lower than some of the yield stresses in references we have cited here[26–28,40,42–46,48] suggests that our experiment might be in an effective inverse Hall-Petch regime (again, using polycrystalline materials as an analogy to the asperity system here). The very limited number of dislocations observed also supports this since the inverse Hall-Petch relationship is theorized to originate in the small grains' inability to generate and collect dislocations near grain boundaries. However, Chandross et al.'s amorphization model provides a theoretical upper bound for the maximum interfacial shear strength which should occur at or near the transition from grain boundary sliding to dislocation-mediated plasticity. Fig. 4 plots the von Mises stress corresponding to this upper bound (green line). The fact that the experimental stresses at yield fall in the range of this theoretical limit suggests that the interface is the "weak link" in the system; the intrinsic strength of

the Ag asperities at small scales is high, and thus sliding may be controlled by interfacial amorphization.

We caution that generalizing these results would require experiments that explore greater overlap distances (i.e., the distance from the top of the lower asperity to the bottom of the upper asperity) so as to promote more bulk plastic activity. Interestingly, Brink and Molinari[51] found using molecular dynamics and finite element simulations that a crossover condition exists for transitioning from bulk plastic flow to interfacial slip. Here, the overlap values range from 17–33% of the effective probe radii for the six experiments. While this could be sufficient to generate bulk plastic flow (and indeed, we observe limited plastic flow in some experiments), predictions like those in[51] serve as motivation for studies over an even wider range of parameter space.

Now we consider the asperity separation process. The experiments had a separation instability at a von Mises stress ranging from 1.28 to 1.81 GPa (1.50 ± 0.20), which corresponds to an effective shear stress ranging from 0.74 to 1.05 GPa (0.87 ± 0.11 GPa) (See Supplemental Figs Table S1). These values are also indicative of high intrinsic material strength. To put these values in context, we consider the two main possibilities for nanoscale contact separation: ductile/atomic necking, and elastic adhesive separation at the pull-off force as predicted by adhesive contact mechanics.

Nanoscale Au-Au asperities have been reported to exhibit ductile necking behavior and even liquid-like behavior during separation; this has been strongly inferred via force and conduction measurements[8,9] and directly observed in TEM studies[28]. Similar behavior has been reported in molecular dynamics simulations[52]. Somewhat similarly, fusing of Ag nanoparticles has been seen in TEM, although the behavior was assisted by the presence of a supporting medium on which Ag atoms could diffuse[53]. Our direct observation using the TEM images with no intervening medium demonstrates that ductile necking is not the dominant mode of separation; as discussed above, plastic deformation after separation is small, except for one of the six experiments as shown in Figs. S9, S12, and S15. As well, a necking instability would normally be manifested as a reduction in the magnitude of the tensile force; this is not seen in any of the experiments. Furthermore, from the force vs. sliding distance plots and from the TEM real-time video, the separation process is observed to involve a sudden instability.

Separation of adhering asperities, even nanoscale in size, is frequently described by adhesive contact mechanics, where the JKR and DMT models provide lower and upper bounds to the pull-off force in pure tensile loading. These models incorporate several assumptions, including homogeneous, isotropic, linear elastic material behavior; axisymmetric paraboloidal asperities with curvature radii that are large compared to the contact width; and purely normal loading (no shear applied). The normal force at separation (the pull-off force, $P_c$) according to the JKR and DMT models are $\frac{3}{2}\pi R_{eff} W$ and $2\pi R_{eff} W$ respectively, where the effective probe radius $R_{eff}^{-1} = R_1^{-1} + R_2^{-1}$ where $R_1$ and $R_2$ are the radii of the two contacting asperities. These are measured directly from the TEM images before contact. $W$ is the work of adhesion; since the crystal orientation of the Ag asperities making contact are not known or controlled, a reasonable estimate can be obtained by assuming the interface is like a grain boundary. Using literature values for orientation-averaged surface[54,55] and grain boundary energies[56,57] of Ag we assume that $W = 2.0$ J/m² as a reasonable estimate (for further information, see Supplementary Discussion 4. The contact's behavior between the JKR and DMT limits is then determined by Tabor's parameter $\mu_T = \frac{R_{eff}^{1/3} W^{2/3}}{E_c^{2/3} z_0}$[58–60],

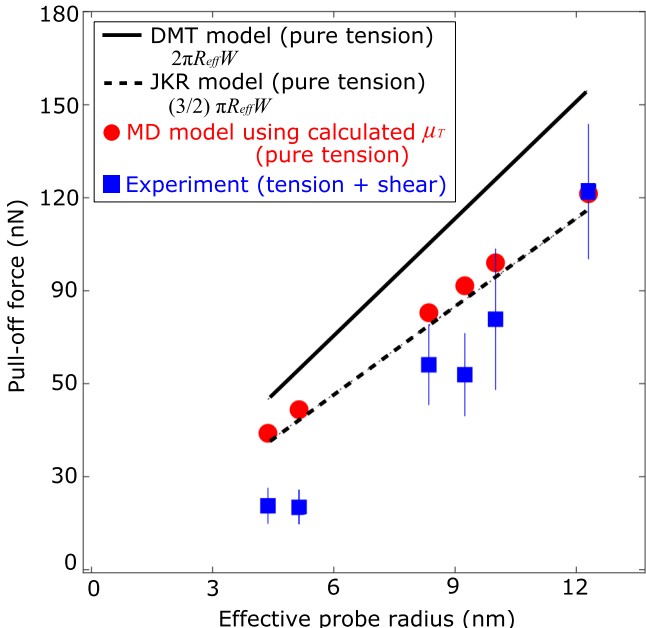

**Fig. 5 The experimental pull-off forces during sliding are well below predictions from adhesive contact mechanics models for separation under pure tension, indicating that shear stresses strongly assist the separation process.** Blue squares: experiments. Red circles: predicted pull-off force from the Maugis-Dugdale model[58] using values of Tabor's parameter μT determined for each experiment. Solid line: predicted pull-off force from the DMT model. Dashed line: predicted pull-off force from the JKR model (see Supplemental Figs Table 2). A value of $W = 2.0$ J/m$^2$ is used for all calculations, as discussed in the text. The experimental pull-off forces are generally well below all of these predicted values, indicating that the applied shear force is playing a role in promoting the separation process. The error bars representing the uncertainty of each experimental value arose from the resolution of the TEM and NEMS actuator, and the calculations are performed as described in Supplementary Discussion 2.

where $E_c = \frac{E}{2(1-\nu^2)}$ (48.1 GPa for Ag), and $z_0$ is the equilibrium separation of the surfaces which is taken to also represent the spatial range of the interfacial forces. We assume $z_0 = 0.2428$ nm, the interplanar spacing of Ag (111) planes, producing values ranging from $\mu_T = 0.83$–1.17 for the six experiments (mean value 1.01). Since $\mu_T < 0.1 \left(\mu_T > 5\right)$ corresponds to the DMT (JKR) limit, our contacts are in between these limits, and the Maugis-Dugdale (MD) model can be used to predict the pull-off force[58–60]. The pull-off forces predicted by the MD model, as well as JKR and DMT limits, are plotted in Fig. 5 (solid and dashed lines) and compared with the values from each experiment. The experimental pull-off forces fall below all predictions, on average only being 69% of the JKR prediction (the lowest bound), showing that these models do not predict the pull-off forces well. However, they do show the same approximately linear increase with $R_{eff}$.

The disagreement with the adhesive contact mechanics predictions for pull-off force and our measurements can potentially be resolved by recognizing that the adhesive contact mechanics models do not consider the effect of the shear stress present due to the sliding. Shear has been incorporated into adhesive contact mechanics by considering mixed-mode loading as first proposed for nanoscale contacts by Johnson[61]. This was further supported by Kim, McMeeking, and Johnson in the limit of small contacts where slip tends to occur concurrently (i.e., without annular pre-slip)[62]. Subsequent work by McMeeking et al[63]. provided a

physical basis for this effect in the JKR limit. They assume that a portion of the mechanical work done by the applied shear force against static friction is stored reversibly. Thus, when released, this energy can help overcome adhesion, reducing the pull-off force from the value obtained in pure tensile loading. This was further supported by Ciavarella and Papangelo[64], whose further analysis showed that the degree of reversibility of this tangential work indeed affects adhesion forces, and does not rely on previously-used assumptions regarding the existence of singularities in the shear stress. Most recently, Peng et al[65]. expanded the analysis beyond the JKR limit to include the full JKR-DMT range, and found good agreement with macroscale experiments they conducted. These models all predict that the applied shear stress reduces the pull-off force, sometimes by appreciable amounts due to the interaction between adhesive and frictional resistance in determining the critical strain energy release rate for separation. The models rely on a parameter known as the shear index $\alpha$ ($0 < \alpha < 1$), which McMeeking et al. showed represents the fraction of the shear energy that is reversibly stored; $\alpha = 0$ means there is no reversible sliding and thus no effect on adhesion, while $\alpha = 1$ represents full reversibility and thus a maximal reduction of adhesion[63]. Experimental data to determine $\alpha$ are limited. In addition to Peng et al.'s experimental results mentioned above, Ciavarella and Papangelo[64] showed that four other literature results could all be reconciled by accounting for different rates of shear loading in the experiments. However, even fewer nanoscale results have been analyzed with this framework. Johnson analyzed prior UHV AFM data[66] for nanoscale Pt tips in sliding contact with muscovite mica with this model, finding $\alpha = 0.2$, corresponding to an 11% reduction in adhesion induced by sliding[61]. Here, we analyze the present data set using Peng et al.'s model. In one case (Experiment 5), we did not resolve any difference between the measured and predicted pull-off force, corresponding to $\alpha = 0$. For the other five experiments, $\alpha$ ranges from 0.52–1.9 (values given in Supplemental Figs Table S2), with an average value of $1.3 \pm 0.8$. Values above 1 are unphysical accordingly to the models; this issue is discussed further below. First, we note that our observations of reduced adhesion are consistent with the aforementioned MD simulations of Ag-Ag nanoasperities, which reported that the contact area reduced by over 40% due to sliding, consistent with our measurements[37].

The variation seen in values of $\alpha$ ranges, including excursions above the limit $\alpha = 1$, could indicate a spurious effect in the measurements. First, it is possible that the reduced pull-off force is due to the work of adhesion being lowered by surface contamination. However, the experiments are performed in ultrahigh vacuum, the Ag asperities were formed by making and breaking the contact prior to experiments (see Methods), and no contamination was observed on the asperities at the atomic level. Moreover, the largest reduction in pull-off force observed (Experiment 2) would require our assumed value of $W$ to be in error by a factor of 2.7. Thus, contamination cannot reasonably explain the ensemble of reduced pull-off force measurements. Second, nanoscale roughness has been shown to cause large reductions in pull-off forces between diamond and diamondlike carbon asperities[67]. However, this effect occurred for observable levels of roughness in TEM measurements, whereas the asperities here are atomically smooth, to within the resolution of the TEM. Thus, nanoscale roughness is not a satisfying explanation for the observed low pull-off forces. Third, it is possible that mechanical vibrations could induce premature separation of the asperities. As stated in Supplementary Discussion 2, the total error in lateral and normal forces, including but not exclusively due to mechanical vibrations, correspond 5–30% of the measured forces at separation, and would be expected to affect the separation force

randomly. Thus, we find it unlikely that the observed reduction is due to purely to spurious vibrations, although this could explain some of the cases where values of $\alpha > 1$ occurred (i.e., the pull-off force reductions were larger than can be explained by the model alone).

Importantly, all of these contact models are for macroscopic contacts. At the nanoscale, continuum mechanics may break down due to atomistic effects[16] such as the effect of nearby free surfaces on the elastic properties, the presence of atomic scale defects like steps or disorder, and effects of finite temperatures. This is a forefront challenge in the theory of contact mechanics, for which the data presented here can potentially be illuminating. In short, our results are the first to report shear-induced reduction of adhesion using direct in situ observations. This indicates that the theoretical concept proposed in the literature, that energy stored during shear displacement can reduce the force of adhesion, is valid for nanoscale Ag-Ag contacts, and potentially a strong effect[63–65]. Importantly, using the classic JKR-DMT models may lead to significant errors when analyzing adhesion data obtained when lateral forces are present.

Finally, we note that sliding asperities may also adhere strongly and lead to fracture away from the interface, leading to wear and debris formation. Aghababaei et al[19]. calculated a critical contact width for this process to occur in shearing asperities based on the increase in the surface energy produced by fracture and the work done by external forces due to shear stress. If the contact width is smaller than the critical width, asperity fracture and wear debris occurs. We calculated the critical width for each of the six experiments, and found that the critical contact widths were 170–1900 times larger than the actual contact width measured in the experiment (See Supplementary Discussion 3). Thus, the model of Aghababaei et al. predicts no asperity fracture, consistent with our results.

In summary, we observed Ag-Ag nanoasperities in sliding contact using a custom-designed NEMS-based apparatus for in situ high resolution TEM observations of contact phenomena in ultrahigh vacuum. The apparatus allows measurement of force and contact geometry, including contact width, of nanocontacts under combined tensile and shear loading in real time. From this, we found that Ag-Ag nanocontacts exhibited multiple surprising and unique phenomena. First, while bulk Ag contacts, like many other ductile metals, easily undergo substantial shear-induced junction growth, the nanocontacts exhibit little to no such behavior despite the applied stresses being in the GPa regime. Second, the nanocontacts are strong. The von Mises stress at observed initial yield points was approximately 20 times higher than that for bulk Ag, reaching more than half of theoretical strength of silver derived from DFT calculations[40], and approaching the ideal strength of Ag based on the commonly-used ideal shear strength of $G/30$. The asperities' strength can be attributed to a low population of pre-existing defects and the difficulty in nucleating and propagating dislocations at small scales within the asperities. Interestingly, the stresses at which yield occurs are consistent with a prediction of interfacial amorphization recently proposed for polycrystalline materials, which occurs in place of dislocation-mediated plasticity at small scales[23]. This, and some evidence from the TEM images themselves, suggest amorphization at the sliding interface as a mechanism for accommodating sliding at small contacts. Third, the asperities separated abruptly during sliding, with separation forces below predictions for either ideal cohesive rupture (concurrent separation, predicted to occur at small scales), or flaw-sensitive rupture (crack propagation-like separation, as described by adhesive contact mechanics models like the JKR model) of asperities, the latter being widely used in the modeling nanocontacts. This discrepancy can be resolved by accounting for the

role of shear stress in promoting adhesive separation, as modeled by Johnson[61] and others. Applying this model to the direct observations of the contacts shows that the coupling effect between shear and adhesion is strong. This indicates that shear stress can reduce the contact area and pull-off force of contacting asperities, leading to higher stresses and earlier tensile rupture than what occurs under pure tension. Thus, applying contact mechanics models like the JKR or DMT model to contacts subjected to shear can lead to significant errors. This in turn may affect predictions for the contact area, friction coefficient, and degree of plasticity in sliding multi-asperity contacts, topics for which reliable predicative methods do not yet exist.

Friction is a complex phenomenon involving multiple simultaneous phenomena, particularly the existence of multiple asperities at the contact interface, the presence of impurities and defects in the materials, and the role of impurities and oxides due to exposure to air. Here, clean (ultra-high vacuum), nanoscale, single-asperity experiments permit observation and deconvolution of specific phenomena without the complicating effects of surface roughness, oxide formation, and contaminant films. The high strength even under combined loading, the lack of junction growth, and the low separation forces in contradiction with standard theories provides foundational information upon which a predictive understanding of frictional interactions can be built with further study.

## Methods

**Ag asperity creation**. A custom-designed NEMS apparatus is illustrated in Fig. 1a, b. The device allows for two displacement axes - in the indentation (load) and the lateral directions - via electrostatically-actuated beams (one for each direction). At the apex of the beams are Ag thin films deposited by thermal evaporation. This produces polycrystalline films with grain sizes of at least 10's of nm. To create a pair of oxide-free asperities, the asperities are brought close together in several ten nm while applying a pulse voltage between the asperities. The flashing procedure made the surfaces purged and oxide removed. This produces a pair of protruding Ag nanoscale asperities (Fig. 1, inset) to examine in the TEM. Each asperity produced in this manner was observed to be crystalline; no grain boundaries were observed within or in the region surrounding the individual asperities. Nominally the two opposing asperities were not cut out of a single crystalline but the roughness formed by the evaporation, could lead to minor misorientation of the crystal axes and other of the two asperities.

**NEMS actuator and transducer**. The electrostatic actuators are integrated into the NEMS device. The device offers the advantages over piezo actuation of increased vibrational and drift stability, linearity, and lack of hysteresis. Forces are calculated by comparing the out-of-contact displacement-actuation voltage response to the in-contact response. The out of contact indentation and lateral displacements are multiplied by the spring constants of the indentation and lateral beams respectively to obtain the force-voltage response. The spring constants of the beams to calculate the forces are obtained experimentally from their resonant frequencies, and the values agree with the analytical and numerical solutions with an accuracy of 1.4%.

**TEM parameters**. The TEM used was a 200 kV HF-2000UHV (Hitachi Ltd., Hitachi-shi, Japan) with a lateral resolution of 0.1 nm and an ultrahigh vacuum (UHV) chamber pressure of $10^{-8}$ Pa ($7.5 \times 10^{-11}$ Torr) at room temperature. The UHV conditions greatly reduce the degree of contaminants adsorbed on the surfaces and present in the gas background of the TEM, in contrast to standard TEM which will have several orders of magnitude greater partial pressures of water, molecular hydrogen, hydrocarbons, and other contaminants. Furthermore, a clean Ag surface was created inside the TEM chamber by approaching the probes with a bias to induce field emission. Videos were captured with a CCD video camera operating at 2 frames/sec. Due to the high stability of the NEMS actuator, atomic resolution is obtained throughout the majority of frames. The effect of the 200 kV electron beam on the Ag sample was considered. The threshold energy of silver is 5 times higher than the energy due to the electron beam of TEM. Therefore, it cannot displace atomic nuclei to interstitial positions and thereby the beam does not degrade the crystalline perfection of the silver part. The current flowing through the Ag sample was in the range of pA, and the temperature increase due to the electron beam was in the range of $10^{-5}$ K (See Supplementary Discussion 1). Therefore, it is reasonable to conclude that electron beam effects are negligibly small.

**Experimental procedure**. Experiments consisted of driving the electrostatic actuator to engage lateral motion of the upper asperity with respect to the lower

one. This led to contact formation, shearing of the upper asperity across the lower one, and then separation. The contact-shear-separation experiment was performed six times. Throughout the measurement, all the frames were extracted from the TEM video. For each frame, the positions of upper and lower asperities were traced to obtain the relative distances in the lateral and vertical direction. Friction, normal, shear and tensile forces were derived from the lateral and vertical displacement. Stress values such as von-Mises stress were calculated from those force values and the width of the actual contact area measured by real-time video of TEM. The detail of uncertainty analysis was described at Supplementary Discussion 2.

## Data availability

Details of the experiments are available within the article and the Supplementary Information. The raw displacement, diameter, and force data, along with the processed shear, normal, and von Mises stress data and code used to generate the processed data can be found at the publicly available repository: https://github.com/zmilne/Ag-in-situ-normal-and-lateral-data.

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

## Acknowledgements
We gratefully acknowledge Prof. R. A. Bernal, Prof. T. Filleter, Prof. Ju Li, Dr. N. Argibay, Dr. M. Chandross, Dr. J. B. McClimon, Prof. Y. Meng, Prof. D. J. Srolovitz, Prof. Qunyang Li, and Prof. K.-S. Kim for useful discussions. TS acknowledges funding from the Japanese society for the promotion of science and the NSK foundation for advancement of mechatronics. TS, ZM, and RWC acknowledge support from the Air Force Office of Scientific Research under grant FA2386-18-1-4083, and from the National Science Foundation under award CMMI-1761874 and CMMI-1854702.

## Author contributions
T.S., Z.B.M., and N.S. provided the conception and design of the study, acquired data Analysis, interpreted the results. M.N. wrote the draft of the manuscript, provided the study materials, laboratory samples, instrumentation, computing resources, and other analysis tools, and acquired the financial support. R.W.C. analyzed and interpreted the data, and revised the manuscript critically for important intellectual content. H.F. managed and coordinated responsibility for the research activity planning and execution.

## Competing interests
The authors declare no competing interests.
