## [Peer Review File · Nature Communications]

Title: Ultrahigh Strength and Shear-Assisted Separation of Sliding Nanocontacts Studied in situREVIEWER COMMENTS

Reviewer #1 (Remarks to the Author):

Please, see PDF attached.

Reviewer #2 (Remarks to the Author):

I found the manuscript very interesting. The authors present an original experimental approach that allows them to investigate the mechanical response of two colliding silver nanoscale asperities. As far as I can tell, the approach is novel and opens a whole new parameter space for experiments. AFM experiments have explored the interaction of a single asperity on a surface, but the interactions of two asperities is largely unexplored and of great topicality and scientific relevance to come up with predictive models for friction.

The experimental method enables to assess the evolution of the contact junction with time (a nice movie is provided in supplemental). Therefore, the authors are able to estimate the adhesive contact area. Using a cleverly designed apparatus, they can also access both the normal force and tangential force. Based on the contact junction angle, they can estimate the Von Mises yield strength. This gives access to the evolution in real time of the stresses and forces as function of contact configuration (including pull-off force). They can also estimate the amount of irreversible plastic deformation once the asperities have slid past one another. The amount of plasticity is surprisingly small, which is an intriguing result. Another intriguing result is that the nanocontacts are very strong, with the estimated Von Mises stress reaching a sizeable portion of the theoretical strength estimated with DFT calculation. This is indeed surprising since the contact junction is made of an imperfect interface between the top and bottom asperities which do not have matching crystallographic orientations. Finally, they compare the pull-off force to theoretical models and find that the shear stress can greatly reduce the contact area and pull-off force of contacting asperities. All these findings should generate healthy discussions in the community.

Overall, I believe the work is novel. I should note that I am not an experimentalist, and other reviewers might confirm the novelty, but to the best of my knowledge I have not seen such data before. The work is original and I believe deserves a high profile journal, and therefore I recommend acceptance of the manuscript.

However, I have a couple of comments that I wish the authors could address or discuss.

1) Part of the methodology is to come up with an estimation of the stress from the force measurements and the contact width. There are two important assumptions behind.

a) First (equation 1), the authors assume they can estimate the Von Mises stress using a plane stress state of stress, i.e. the stress tensor is essentially a 2x2 matrix. This assumption is valid for thin

structures (think of a wide plate that is very thin in the third direction, and with all loadings in plane). Plane strain is another assumption that is often used to reduce a 3D to a 2D problem. Plane strain would be valid for thick structures in the third dimension. The contacting nanoscale asperities fall in neither category. They are not thick (cylinders) nor thin (penny shape), they are in fact 3D objects with quasi hemi-spherical shapes. So this begs the question how accurate the 2D plane stress assumption is. A 3D finite-element calculation with plasticity might clarify this.

b) Second, the contact area is assumed to be a perfect disk (equation 4) and not a line, which again shows that it is not in a plane stress problem. The asperities are not perfect hemispheres and appear bumpy, so maybe the contact area is elliptical. How accurate is the estimation of the contact area with equation 4.

a) and b) taken together introduce some uncertainty on the estimation of the Von Mises stress, considering that the apparent high strength of these nanoscale silver asperities (figure 4) is a key finding of the manuscript, I believe it is important to address/discuss those uncertainties.

2) The authors have explored an interesting parameter space, but I wonder if it could be further enlarged. Let us define the overlap by the vertical length over which the asperities establish contact. A lot of the observations are conditioned by the rather small overlap between the colliding asperities, see figure 1 c i. Is the methodology robust to probe larger overlaps? Larger overlaps would result in larger contact junctions and forces, which might lead to failure of the apparatus, so might not be doable. In any case, I wondered if the limited junction growth observed in these experiments would not be much more prominent for more overlap. It seems that the configuration that has been probed leads to more slip at the contact interface than to bulk plastic activity. This finding might be very specific to the probed geometry, and perhaps the authors should not generalize too much their findings, at least before they confirm them for a broader space of geometries.

Along the same line, the discussion line 234-241 on the difficulty of differentiating between dislocation activity and slip might be clarified by changing the attack angle of the collision and/or the overlap. Low contact angles favor slip along the interface whereas high angles favor plastic activity. The authors might be interested in the discussion in Brink et al., Phys. Rev. Mat., 2019, that provides a crossover angle for transitioning to slip at the interface to plastic flow.

3) Finally, I was a bit confused by the discussion on the pull off force. If the shear stress have for effect to reduce the contact area, then shouldn't it be automatic that the pull off force will reduce in proportion? My interpretation is that the shear load leads to a progressive reduction of the contact area (a small neck is formed, although very limited spatially), and therefore a smaller pull off force than expected. I guess I have a hard time understanding why this is a surprise if we account for the whole history of deformation and reduction of contact area. Thank you for clarifying.

Minor comments:

Legend fig 5, W symbol is missing in legend of DMT and JKR model.

Line 315, please introduce what is " a " in Persson's model.

I encourage the authors to take the manuscript through a thorough proofreading, I could spot a number

of typos. Also, I believe that the supplemental file that has been provided with the manuscript corresponds to prior iteration among the authors instead of to the definitive version. Please, make sure to attach the final version next time.

Non exhaustive list of typos:

Line 37: ``metal contacts are important \textit{as}..."

Line 38: ``and are \textit{also} important ..."

Line 43: there is a dot missing between the reference and ``While the contact..."

Line 156: ``small relative amount \textit{of} ..."

Line 304: ``show that these model\textit{s} ..."

Line 345: the reference to Aghabebai et al. should be [19], according to the reference list provided with the manuscript. This misalignment is also present in the Supplementary Material file.

Line 348: ``wear debris \textit{occurs} ..."

Line 365, ``This, \textit{an} some..."

Throughout the manuscript please use text instead of numbers for numbers smaller than 12. Example:

Line 138 "The other 3..."

Reviewer #3 (Remarks to the Author):

The paper presents precise measurements of shear resistance of nano-size Ag-Ag asperities and atomic TEM pictures/videos of cross section of the deformation of the asperities during sliding. Comparing with the previous experimental results reported by the authors, this work covers a wider range of asperity size (tip radius from ~5nm to ~20nm), and better resolution of contacting region. A mostly important finding of the work is the ultrahigh strength of the shear resistance of silver, which has been discussed in detail with respect to the size effect of yield strength of polycrystalline metals reported by other research groups. The results are fundamental for better understanding of the mechanisms of adhesion, friction and wear of metals at the nanoscale. Some minor revisions are suggested as below.

1) The value of work of adhesion W . The authors assume that W equals to 2.52J/m^2 , twice surface energy of silver, in line 297, page 12. This is probably not true. Work of adhesion W is defined as the external work required to separate a junction formed between two clean solid surfaces of a and b in the direction normal to the interface, or $W = \gamma_a + \gamma_b - \gamma_{ab}$, γ_a and γ_b are the surface energy of surface a and surface b respectively, and γ_{ab} is the interfacial energy. The surface energy of single crystalline metals depends on the crystallography plane of the surface, the surface energy of (100) plane differs with those of (110) and (111) planes for a FCC metal like silver. The value of γ_{ab} is zero only when surface a contacts with surface b on the exact same crystalline plane and aligns with each other without any misfits of orientation. Otherwise, γ_{ab} does not disappear. For the friction tests of Ag-Ag contacts studied under TEM in this work, The value of W , 2.52J/m^2 used in the pull-off force estimations, Fig.5, based on JKR and DMT models, is probably overestimated.

2) It is hard to find which crystalline planes, (100), (110) or (111), are in contact at initial, and if there is

a misalignment between the contacting surfaces, and any changes during the asperity sliding tests, from the TEM videos and pictures provided. These details are important for in-depth understanding of the experimental findings.

3) Some literatures on nanowire tensile measurements of yield stress and inverse Hall-Petch size effect are cited in discussions to explain the ultrahigh shear strength of the nanocontacts in sliding. However, yield stress of metals only expresses the bulk property of the tested samples. In sliding experiments, not only bulk properties but also surface properties of materials involved in the shear resistance. In his book "Friction and Wear of Materials", E.Rabinowicz has pointed out that surface energy plays more and more important role in tensile and hardness tests when the sample or indenter size is down to a few micrometers. For the nanoscale asperities investigated in the study, contributions of surface effect to shear resistance are not negligible.

Point to Point Response to the Reviewers
Manuscript ID: NCOMMS-21-31426A-Z

We sincerely appreciate the editor and reviewers' valuable and insightful comments on our manuscript. We have revised and improved our manuscript as suggested. In the following, we provide a point-to-point response to every question raised. We have highlighted portions of the manuscript that have been updated in response to the reviewers' comments.

Reviewers' comments:

Reviewer #1 (Comments for the authors):

The authors have performed adhesion tests of Ag nanocontacts in presence of shear loads. Their sophisticated experiments provide for the first time in situ measurements of both normal and shear forces, and also access to the contact width. This is achieved through a dedicated apparatus (designed on purpose) exploiting nanoelectromechanical components. Measurements are performed in ultrahigh vacuum for nanometric, almost paraboloid protrusions of Ag, with characteristic size down to few nanometers. Using nanocontacts have several advantages, which are clearly explained along the paper.

Response: We thank the reviewer for these very positive comments on the work.

I will support the paper to be published in Nature Communications only following major revisions in the interpretation of the data.

Issue 1-1. *My major concern regards the analysis the authors made using the model by Johnson [57] (which is an old one, see later). Johnson model is based on LEFM. It basically states the balance between the energy release rate due to normal and tangential loads and an **effective** work of adhesion G_c . This is an "effective" one, as it is known from mixed-mode fracture mechanics that in mixed-mode conditions interfaces are usually more tough than under pure mode I conditions. So far, there is no agreement on which model to be used for this "effective" toughness, and Johnson proposed (this is not physics based) a linear model where $G_c = w(1 + \alpha * (G_{II}/G_I))$. Although arbitrary, this is a convenient form as **by varying alpha from 0 to 1**, one obtains to clear limits. $\alpha = 0$ means the interface does not get stronger under mixed-mode. This is the case where one gets the stronger reduction of contact area, the stronger effect of shear tractions, as in the seminal work of Savkoor and Briggs (1977). The other limit is that of $\alpha = 1$, which means the interface is insensitive to shear tractions, hence the model goes back to the classical JKR model for normal contact.*

It is concerning to me the authors have found $\alpha = 2.4 \pm 0.8$ and conclude that Johnson [57] model is appropriate for describing these nanocontacts adhesive experiments. As far as I understand, Johnson [57] model is completely inaccurate in doing so. One reason this happens may be related to the fact that Johnson model, without shear, would be the JKR classical model, which cannot be the correct model if the authors state the Tabor parameter to be about 0.57-0.81.

In my opinion, this could invalidate all the authors' analysis, unless they have an explanation for it I did not grasp. Nevertheless, their measurements are a step forward in tribology; hence, the paper may be still worth Nat Comm if appropriate amendments on the data interpretation will be made at the next step.

Response: We thank the reviewer for these comments. We completely agree that since our calculated value of Tabor's parameter for our system reported in the original submission fell between 0.57 – 0.81, the JKR model is not appropriate to describe the contact mechanics (we have updated these values but the conclusion remains the same; Tabor's parameter indicates we are in the transition regime between JKR and DMT). It is for this reason that, in the original submission, we calculated and presented results using both the JKR and DMT limits for analyzing the relationship between the work of adhesion and the probe radii, as presented in Fig. 5, to illustrate the range of possible values. We note that our calculation of Tabor's parameter requires knowledge of several parameters including the work of adhesion, and the equilibrium separation of the surfaces, which are not known *a priori*, making estimation of Tabor's parameter approximate. We have clarified this point on page 14, 15, 16, and 17 of the manuscript. Note that, in response to Reviewer 3, we have modified the value used for the work of adhesion, as discussed further below.

We also fully agree that we should have applied an updated model for interpreting the effect of shear on the contact, and that the Johnson model in *Continuum mechanics modelling of adhesion and friction*. **Langmuir** 12, 4510-4513 (1996) (reference [57] in our originally-submitted manuscript) is only appropriate for JKR contacts. In fact, we made an error in citing that paper, as we actually meant to instead cite his subsequent paper, *Adhesion and friction between a smooth elastic spherical asperity and a plane surface*. Proc. Roy. Soc. London A 453, 163-79 (1997). In that 1997 paper, Johnson does consider non-JKR cases by varying Tabor's parameter, and it is that analysis which we actually applied in our original submission. We note that we did correctly cite the paper by Kim, McMeeking, and Johnson, K.L. *Adhesion, slip, cohesive zones and energy fluxes for elastic spheres in contact*. **J. Mech. Phys. Sol.** 46, 243-66 (1998) (reference [58] in our originally-submitted manuscript), as that paper clarifies that Johnson's 1997 paper should only apply in the limit of small contacts where slip tends to occur without annular pre-slip; this limit is appropriate for the very small contacts we are studying here.

Regardless, we fully agree the more recent analysis of the interaction between friction and adhesion should be used since, as the reviewer points out, a more general and updated approach is available. We have thus removed original reference [57], and instead apply the much more recent and comprehensive analysis of Peng *et al.* (2021), as discussed in the next point. We greatly appreciate the reviewer pointing out the existence of this recent model and the related literature, which has been very helpful. This issue is further discussed below.

Issue 1-2. *Johnson model [57] is a hold one. There have been many papers published on this theme recently, both numerical and experimental contributions, involving macro contacts. There is no trace of this. A non-exhaustive list is reported hereafter, where I have restricted myself only to the references published in the last 3 years...*

- [1] Sahli, R., Pallares, G., Ducottet, C., Ali, I.B., Al Akhrass, S., Guibert, M., Scheibert, J.: Evolution of real contact area under shear and the value of static friction of soft materials. *Proc. Natl Acad. Sci. U.S.A.* 115(3), 471–476 (2018)
- [2] Ciavarella, M.: Fracture mechanics simple calculations to explain small reduction of the real contact area under shear. *Facta Univ. Ser. Mech. Eng.* 16(1), 87–91 (2018)
- [3] Papangelo, A., Ciavarella, M.: On mixed-mode fracture mechanics models for contact area reduction under shear load in soft materials. *J. Mech. Phys. Solids* 124, 159–171 (2019)
- [4] Papangelo, A., Scheibert, J., Sahli, R., Pallares, G., Ciavarella, M.: Shear-induced contact area anisotropy explained by a fracture mechanics model. *Phys. Rev. E* 99(5), 053005 (2019)
- [5] Mergel, J.C., Sahli, R., Scheibert, J., Sauer, R.A.: Continuum contact models for coupled adhesion and friction. *J. Adhes.* 95(12), 1101–1133 (2019)
- [6] Sahli, R., Pallares, G., Papangelo, A., Ciavarella, M., Ducottet, C., Ponthus, N., Scheibert, J.: Shear-induced anisotropy in rough elastomer contact. *Phys. Rev. Lett.* 122(21), 214301 (2019)
- [7] McMeeking, R.M., Ciavarella, M., Cricri, G., Kim, K.S.: The interaction of frictional slip and adhesion for a stiff sphere on a compliant substrate. *J. Appl. Mech.* 87(3), 031016 (2020)
- [8] Das, D., Chasiotis, I.: Sliding of adhesive nanoscale polymer contacts. *J. Mech. Phys. Solids* 103931, 140 (2020).
- [9] Lengiewicz, J., de Souza, M., Lahmar, M.A., Courbon, C., Dalmas, D., Stupkiewicz, S., Scheibert, J.: Finite deformations govern the anisotropic shear-induced area reduction of soft elastic contacts. *J. Mech. Phys. Solids* (2020).
- [10] Mergel, J.C., Scheibert, J., Sauer, R.A.: Contact with coupled adhesion and friction: computational framework, applications, and new insights. *arXiv preprint. arXiv :2001.06833* (2020)
- [11] Peng, B., Li, Q., Feng, X. Q., & Gao, H. (2021). Effect of shear stress on adhesive contact with a generalized Maugis-Dugdale cohesive zone model. *Journal of the Mechanics and Physics of Solids*, 148, 104275.
- [12] Ciavarella, M., & Papangelo, A. (2020). On the degree of irreversibility of friction in sheared soft adhesive contacts. *Tribology Letters*, 68(3), 1-9.

Response: We thank the reviewer for pointing out this issue and providing a comprehensive set of references. We were not aware of all of these advances, which are generally very recent. Note, as mentioned above, we intended to cite (and actually used the analysis of) Johnson's 1997 work in **Proc. Roy. Soc.**, and also cited the follow-on work of McMeeking, Kim and Johnson in 1998 in **J. Mech. Phys. Sol.**

Due to the strict limitations of the journal, we cannot include all these references in the main text. We have put the ones we deem to most relevant in the main text as we find these papers to be of very high quality and well worth citing and (briefly, due to space limits) discussing:

[1] McMeeking, R.M., Ciavarella, M., Cricri, G., Kim, K.S. (2020). *The interaction of frictional slip and adhesion for a stiff sphere on a compliant substrate*. **J. Appl. Mech.** 87, 031016.

[2] Peng, B., Li, Q., Feng, X. Q., & Gao, H. (2021). *Effect of shear stress on adhesive contact with a generalized Maugis-Dugdale cohesive zone model*. **J. Mech. Phys. Sol.**, 148, 104275.

[3] Ciavarella, M., & Papangelo, A. (2020). *On the degree of irreversibility of friction in sheared soft adhesive contacts*. **Tribol. Lett.**, 68, 1-9.

These changes are found on page 14.

Further comments below about applying these models to our data are provided in the next point.

Issue 1-3. *Apart from acknowledging the work of others in this field, I find particularly interesting the work of McMeeking et al (2020). The idea put forward here, is that we may finally have a physics based explanation of the behavior of interfaces under mixed-mode conditions. McMeeking et al (2020) speculate that it is not the interface that gets tough in mixed-mode conditions, instead in the balance one has to equate the reversible energy release rate (ERR) with the work of adhesion. This makes the difference hence in tangential shearing only a tiny part of the mode II ERR can be reversibly retrieved (see McMeeking et al. (2020)), which may explain why shear tractions may up to a different degree weaken adhesion. A shear-index was defined as $\alpha = G^{rev}/G \leq 1$ (or λ in some works). Following McMeeking et al. (2020), Ciavarella and Papangelo (2020) and Peng et al (2021) have given precious contributions. Ciavarella and Papangelo (2020) have shown that α , when obtained from Literature results, seems to scale as a power law with the loading velocity. Peng et al (2021) have developed a MD contact model accounting for a shear tractions. They borrow the idea of McMeeking et al. (2020) and have found in their experiments $\alpha = 1$ for soft and $\alpha = 0.43$ for stiff contacts.*

What about this paper results? What would be the shear-index in these Ag-Ag nanocontacts experiments? I believe Peng et al (2021) model is exactly what the authors should look at...

Response: We thank the reviewer's insightful comments on our work in the context of this literature. We have reviewed this literature and find the work to be impressive and indeed relevant to our experiments. We have reviewed these models, and found an average value of $\alpha = 1.3 \pm 0.8$. While the spread is large and the mean is outside the physically acceptable range, this is significantly more realistic than our previous calculations. In the revised manuscript, we discuss factors which may contribute to spread and overestimates in the values of α obtained.

We agree with the reviewer's comment that all of the models are worked out for macroscopic contacts. Thus, we also discuss the fact that the continuum model may need to be applied with caution for these nanoscale contacts considering the contact radii here are in the range of 4-12 nm. At the nanoscale, caution must be exercised not only due to the small size itself but since continuum mechanics could break down due to atomistic effects, as elegantly shown by Robbins and co-workers (Luan, B. and Robbins, M.O. (2005) *The breakdown of continuum models for mechanical contacts*. **Nature** 435, 929-32), along with many others. Possible origins of this breakdown include the effect of nearby free surfaces on the elastic properties of the Ag asperities; atomic scale defects like steps or disorder; and the effect of finite temperatures which can lead to thermal fluctuations, diffusion, and other thermally-activated processes.

We have addressed these points in the main manuscript. While further work is clearly needed to understand this fully, including the spread in values of α , such an undertaking would be its own research effort. Addressing this is beyond the scope of the present paper. Our primary results focus on the high intrinsic strength and the decohesion behavior of metal contacts under combined loading, which are novel. The portion of the results pertaining to the coupling of adhesion and friction are shared here as they provide an opportunity to further advance the discussion of this interesting and important phenomenon, with a particular focus on possible

nanoscale effects at play. The results point toward strong coupling of adhesion and friction for sheared adhesive nanocontacts, and that nanoscale effects may be at play, which requires modeling that takes atomistic detail into account.

Issue 1-4. *I do not agree with the sentence (line 306) “these models treat the separation process as an analog to fracture”. This is certainly true for JKR, not true for DMT. Please see Ciavarella, M., Joe, J., Papangelo, A., & Barber, J. R. (2019). The role of adhesion in contact mechanics. Journal of the Royal Society Interface, 16(151), 20180738, for a clear explanation of the differences between DMT and JKR.*

Response: The statement in that sentence was motivated by the results of Maugis, who obtained the full transition from JKR to DMT using mode I fracture mechanics applied to a Dugdale crack (Maugis, D., *Adhesion of spheres: the JKR-DMT transition using a Dugdale model. J. Colloid Interface Sci.* 150, 243 (1992).). We agree that the analogy to fracture essentially disappears in the DMT limit. In light of the next point discussed below, we have removed the discussion of this point as it is no longer relevant; we have also cited the paper of Ciavarella *et al.* mentioned above to provide further clarification.

Issue 1-5. *Page 13. I think the authors should better clarify in the text that JKR is the “flaw-sensitive” and Persson is the “flaw-insensitive” regime. Try just to make the text more clear.*

Response: We agree that this wording was unclear. Moreover, while there is an interesting discussion to consider regarding the issue raised in the work of Persson, B. (2003) *Nanoadhesion. Wear* 254, 832 (Ref. 56 in the original submission), we have conducted further analysis and found that Persson’s analysis, which is for a rigid flat punch, is not as relevant for an elastic sphere-plane contact. This is because the length scale at which his predicted transition from crack-like separation to uniform rupture occurs at an extremely small scale (< 3 nm). We believe this discussion distracts from the larger points of the paper and is of little consequence, and hence have removed it from the manuscript.

Issue 1-6. Line 301. I think it should read “DMT (JKR)”

Response: The referee is correct; this was an error. We thank the referee for pointing it out. We have corrected it.

Reviewer #2 (Comments for the authors):

I found the manuscript very interesting. The authors present an original experimental approach that allows them to investigate the mechanical response of two colliding silver nanoscale asperities. As far as I can tell, the approach is novel and opens a whole new parameter space for experiments. AFM experiments have explored the interaction of a single asperity on a surface, but the interactions of two asperities is largely unexplored and of great topicality and scientific relevance to come up with predictive models for friction.

The experimental method enables to assess the evolution of the contact junction with time (a nice movie is provided in supplemental). Therefore, the authors are able to estimate the adhesive contact area. Using a cleverly designed apparatus, they can also access both the normal force and tangential force. Based on the contact junction angle, they can estimate the Von Mises yield strength. This gives access to the evolution in real time of the stresses and forces as function of contact configuration (including pull-off force). They can also estimate the amount of irreversible plastic deformation once the asperities have slid past one another. The amount of plasticity is surprisingly small, which is an intriguing result. Another intriguing result is that the nanocontacts are very strong, with the estimated Von Mises stress reaching a sizeable portion of the theoretical strength estimated with DFT calculation. This is indeed surprising since the contact junction is made of an imperfect interface between the top and bottom asperities which do not have matching crystallographic orientations. Finally, they compare the pull-off force to theoretical models and find that the shear stress can greatly reduce the contact area and pull-off force of contacting asperities. All these findings should generate healthy discussions in the community.

Overall, I believe the work is novel. I should note that I am not an experimentalist, and other reviewers might confirm the novelty, but to the best of my knowledge I have not seen such data before. The work is original and I believe deserves a high profile journal, and therefore I recommend acceptance of the manuscript.

Response: We thank the reviewer for these very positive comments on the work.

However, I have a couple of comments that I wish the authors could address or discuss.

Issue 2-1. Part of the methodology is to come up with an estimation of the stress from the force measurements and the contact width. There are two important assumptions behind.

a) First (equation 1), the authors assume they can estimate the Von Mises stress using a plane stress state of stress, i.e. the stress tensor is essentially a 2x2 matrix. This assumption is valid for thin structures (think of a wide plate that is very thin in the third direction, and with all loadings in plane). Plane strain is another assumption that is often used to reduce a 3D to a 2D problem. Plane strain would be valid for thick structures in the third dimension. The contacting nanoscale asperities fall in neither category. They are not thick (cylinders) nor thin (penny shape), they are in fact 3D objects with quasi hemi-spherical shapes. So this begs the question how accurate the 2D plane stress assumption is. A 3D finite-element calculation with plasticity might clarify this.

b) Second, the contact area is assumed to be a perfect disk (equation 4) and not a line, which again shows that it is not in a plane stress problem. The asperities are not perfect hemispheres and appear bumpy, so maybe the contact area is elliptical. How accurate is the estimation of the contact area with equation 4.

a) and b) taken together introduce some uncertainty on the estimation of the Von Mises stress, considering that the apparent high strength of these nanoscale silver asperities (figure 4) is a key finding of the manuscript, I believe it is important to address/discuss those uncertainties.

Response: The reviewer raises valid points. Regarding point (a), we agree that the plane stress assumption was used, i.e.,

$$S_{vM} = \sqrt{\sigma_{\square}^2 + 3\tau_{\square}^2}$$

where σ and τ are the mean normal and shear stress values at the points of interest, as reported in Table S1. For the pure plane strain case, the von Mises stress are in fact higher, and are given by

$$S'_{vM} = \sqrt{((1 + \nu)\sigma)^2 + 3\tau_{\square}^2}.$$

Using $\nu_{Ag} = 0.365$, the resulting von Mises stresses in Table S1 are 10-33% higher than for the plane stress case, rendering our measured values even closer to the ideal strength. Our use of the plane stress calculation is thus a conservative (lower bound) approach. Note that one can rigorously calculate the location and value of the largest von Mises stress for the inhomogeneous stress distribution in a Hertz, DMT, or JKR contact analytically. This location resides below the interface for the case of pure normal loading (no applied shear), and for the Hertz model is equal to 0.93σ , rather close to the plane stress value of $S_{vM} = \sigma$ (since $\tau = 0$). Adding in a shear stress of approximately 0.5σ (this is, on average, approximately the size of our values of τ) leads to a von Mises stress of approximately 1.3σ according to the calculations of Wang, Q.J. and Zhu, D., Hertz Theory: Contact of Spherical Surfaces, in Encyclopedia of Tribology. (eds. Q.J. Wang & Y.-W. Chung) 1654-1662 (Springer US, Boston, MA; 2013). In comparison, the plane stress value is 1.32σ . In other words, the plane stress calculation for the von Mises stress using mean stress values tends to agree rather well with the actual value of the largest von Mises stress. Unfortunately, a full finite element model may not shed more light as it would require making assumptions about the unknown interfacial traction-separation behavior in addition to the unknown effect of shear-normal coupling, which is the topic discussed with Reviewer 1. Considering this, we do not expect such calculations to provide insights within the limits of our experimental uncertainty. However we fully agree with the Reviewer that this issue is a limitation that must be acknowledged. We have thus modified the text on page 5 to make clear that we are using mean stress values and assuming plane stress in our calculated values.

Regarding point (b), this is an intrinsic limitation of the TEM technique, since it only provides effectively a plane (2D) view of the contact; without a so-called double-tilt TEM holder, we cannot vary the viewing direction. Thus, we and others commonly use the working assumption that the contact is sufficiently close to axisymmetric so that the subsequent analysis is reasonable. Other published nanoscale sliding contact experiments with subsequent 3-D reconstruction of the asperities have shown that axisymmetric asperities retain axisymmetry to a large degree despite undergoing substantial sliding and wear (for example, Liu, J., Notbohm, J.K., Carpick, R.W. and Turner, K.T. *Method for Characterizing Nanoscale Wear of Atomic Force Microscope Tips*. *ACS Nano* 4, 3763–3772 (2010)), but admittedly, there are only a few such examples in the literature. In our original submission, we did explicitly mention that axisymmetry was an assumption, but we have now updated the manuscript to make the use of this assumption, and the reason for it, clearer, as discussed on page 5. Unfortunately, it is not possible to pin down this question further due to fundamental limitations of the experimental system. Future innovations in instrumentation may permit this issue to be addressed, and the issue could be considered with future atomistic simulations which we hope our study may motivate.

*Issue 2-2. The authors have explored an interesting parameter space, but I wonder if it could be further enlarged. Let us define the overlap by the vertical length over which the asperities establish contact. A lot of the observations are conditioned by the rather small overlap between the colliding asperities, see figure 1 c i. Is the methodology robust to probe larger overlaps? Larger overlaps would result in larger contact junctions and forces, which might lead to failure of the apparatus, so might not be doable. In any case, I wondered if the limited junction growth observed in these experiments would not be much more prominent for more overlap. It seems that the configuration that has been probed leads to more slip at the contact interface than to bulk plastic activity. This finding might be very specific to the probed geometry, and perhaps the authors should not generalize too much their findings, at least before they confirm them for a broader space of geometries. Along the same line, the discussion line 234-241 on the difficulty of differentiating between dislocation activity and slip might be clarified by changing the attack angle of the collision and/or the overlap. Low contact angles favor slip along the interface whereas high angles favor plastic activity. The authors might be interested in the discussion in Brink *et al.*, *Phys. Rev. Mat.*, 2019, that provides a crossover angle for transitioning to slip at the interface to plastic flow.*

Response: We appreciate the Reviewer's suggestion. Thanks to the high spatial resolution of our technique permits us to calculate the overlap values; while the overlaps are small, the asperity radii are small as well. Accordingly, the overlap ranges from 17-33% of the effective probe radii for the six experiments, these values are not negligible, although we agree that even larger overlaps would be interesting to study in the future. The Reviewer's comment about generalization is a valid one and so we have added context on page 11 and 12 to address this point. The paper by Brink *et al.* is indeed very interesting and we have added a citation to that work to highlight the fact that other regimes of behavior could be accessed with further experiments. Unfortunately, such studies are beyond the scope of the present work, but we appreciate the Reviewer's comment as it has guided us to further discuss the point raised.

Issue 2-3. Finally, I was a bit confused by the discussion on the pull off force. If the shear stress have for effect to reduce the contact area, then shouldn't it be automatic that the pull off force will reduce in proportion? My interpretation is that the shear load leads to a progressive reduction of the contact area (a small neck is formed, although very limited spatially), and therefore a smaller pull off force than expected. I guess I have a hard time understanding why this is a surprise if we account for the whole history of deformation and reduction of contact area. Thank you for clarifying.

Response: The reviewer is correct that, according to the models cited, the shear force can induce a reduction in contact area (due to slip of an annular region around the remaining contact area) and a concomitant reduction in pull-off force. Unfortunately, this can be very difficult to observe because (a) the reduction in contact area can involve only a minute increase in the gap between the solids in the annular region of slip; and (b) while a fraction of this reduction can exist stably, near pull-off further reduction occurs unstably. Elaborate in-situ methods have been used to observe this effect at the macroscale for soft materials where the effect is large and the dynamics slow, e.g., see references 9, 10, and 11 in the comment of Reviewer #1, Issue 1-2 (these papers also further expand on the robust debate that exists about the degree of shear-induced weakening of adhesive contacts). Unfortunately, at the nanoscale, despite the high resolution of the TEM, the effect is extremely difficult to capture in live images.

Legend fig 5, W symbol is missing in legend of DMT and JKR model.
Line 315, please introduce what is “ a ” in Persson’s model.

Response: We thank the reviewer for these comments. We have rectified the first error. Regarding the second one, as explained above we have removed the discussion of Persson’s model.

Non exhaustive list of typos:

Line 37: “metal contacts are important **as**...”

Line 38: “and are **also** important ...”

Line 43: there is a dot missing between the reference and “While the contact...”

Line 156: “small relative amount **of** ...”

Line 304: “show that these models ...”

Line 345: the reference to Aghabebei et al. should be [19], according to the reference list provided with the manuscript. This misalignment is also present in the Supplementary Material file.

Line 348: “wear debris **occurs** ...”

Line 365, “This, **an** some...”

Throughout the manuscript please use text instead of numbers for numbers smaller than 12. Example: Line 138 “The other 3...”

Response: We thank the reviewer for their careful review. We have corrected all of these errors, and reviewed the manuscript fully.

Reviewer #3 (Comments for the authors):

The paper presents precise measurements of shear resistance of nano-size Ag-Ag asperities and atomic TEM pictures/videos of cross section of the deformation of the asperities during sliding. Comparing with the previous experimental results reported by the authors, this work covers a wider range of asperity size (tip radius from ~5nm to ~20nm), and better resolution of contacting region. A mostly important finding of the work is the ultrahigh strength of the shear resistance of silver, which has been discussed in detail with respect to the size effect of yield strength of polycrystalline metals reported by other research groups. The results are fundamental for better understanding of the mechanisms of adhesion, friction, and wear of metals at the nanoscale. Some minor revisions are suggested as below.

Response: We thank the reviewer for these very positive comments on the work.

Issue 3-1. The value of work of adhesion W . The authors assume that W equals to 2.52J/m^2 , twice surface energy of silver, in line 297, page 12. This is probably not true. Work of adhesion W is defined as the external work required to separate a junction formed between two clean solid surfaces of a and b in the direction normal to the interface, or $W = \gamma_a + \gamma_b - \gamma_{ab}$, γ_a and γ_b are the surface energy of surface a and surface b respectively, and γ_{ab} is the interfacial energy. The surface energy of single crystalline metals depends on the crystallography plane of the surface, the surface energy of (100) plane differs with those of (110) and (111) planes for a FCC metal like silver. The value of γ_{ab} is zero only when surface a contacts with surface b on the exact same crystalline plane and aligns with each other without any misfits of orientation. Otherwise, γ_{ab} does not disappear. For the friction tests of Ag-Ag contacts studied under TEM in this work, The value of W , 2.52J/m^2 used in the pull-off force estimations, Fig.5, based on JKR and DMT models, is probably overestimated.

Response: We thank the reviewer for this comment. We fully agree with the reviewer’s point that 2.52J/m^2 is an upper bound, as it based on the ideal case of perfect rotation alignment of two identical crystal planes. Indeed, elsewhere in the original manuscript we stated, “We now consider the contact of the two asperities in the context of how polycrystalline metals deform; essentially, the contact is a bicrystal interface with proximal surfaces,” in agreement with the reviewer’s point.

While perfect alignment would be extremely unlikely for the case of two randomly oriented asperities, we do note that our asperities made contact several times first, and so their surfaces were exposed after slow sliding

and fracture. It is thus possible that lower energy (*i.e.*, low index) surfaces are more likely to be exposed, as the reviewer refers to in their next point. Unfortunately, our TEM images, as we discuss below, do not permit unambiguous determination of the crystal orientation of the two asperities' surfaces. Thus, it is indeed appropriate to reconsider the value chosen for the work of adhesion W .

It is therefore better to use orientation-averaged values, as the Reviewer suggests. When two different crystal surfaces 1 and 2 of the same material form an interface, the work required to separate them is expressed as $W = \gamma_1 + \gamma_2 - \gamma_{GB}$, where γ_{GB} is the grain boundary energy. We have carefully searched the literature and determined that reasonable estimated values of the respective energy terms are $\gamma_1 = \gamma_2 = 1.3 \text{ J/m}^2$ (as this is an orientation-averaged value for Ag) and $\gamma_{GB} = 0.6 \text{ J/m}^2$, leading to $W=2.0 \text{ J/m}^2$.

We have updated the manuscript on page 13 accordingly, and discuss the justification for this choice of W in Supplementary Discussion 4. Calculations that depend on W have been recalculated, and we have found the results are in better agreement with models as discussed in the reply Reviewer 1's comments. We appreciate this suggestion by the Reviewer, which has helped us improve the manuscript.

Issue 3-2. *It is hard to find which crystalline planes, (100), (110) or (111), are in contact at initial, and if there is a misalignment between the contacting surfaces, and any changes during the asperity sliding tests, from the TEM videos and pictures provided. These details are important for in-depth understanding of the experimental findings.*

Response: Since Ag was deposited on the probes using an evaporator, numerous grains, several tens of nm wide, are deposited. The TEM images show mostly striped patterns; we occasionally see regions of limited size that resolve individual atomic columns (which appear as isolated dots in the TEM image). Regardless, without optimizing via double tilting of the TEM specimen and/or the use of diffraction (neither of which were available for this experiment), it is not possible to unambiguously determine the crystal orientation of the surfaces of the two asperities.

We note that our experiments are distinct from sliding at a single crystal interface, because the friction occurs at an interface between two distinct crystal planes, as the reviewer states. Furthermore, the rounded asperity's motion involves tracing over the other rounded asperity. Therefore, the two crystal orientations at the interface are constantly changing as sliding occurs. It is thus impossible, if not incorrect, to specify a single description of the interfacial structure. Rather, as mentioned in the previous comment, we use the orientation-averaged surface and grain boundary energies of Ag as a guide.

Note that, during each sliding experiment, we did not see any evidence of dislocation nucleation or motion, nor was there any rotation of the crystalline since the orientation of the resolved stripes and columns never changed. Thus, there was no change in the asperity grain itself. It is therefore valid to conclude that the experiments primarily involve sliding at the contact interface, with very limited plastic deformation (as quantified in the manuscript).

We have made these points clearer in the manuscript on pages 13.

Issue 3-3. *Some literatures on nanowire tensile measurements of yield stress and inverse Hall-Petch size effect are cited in discussions to explain the ultrahigh shear strength of the nanocontacts in sliding. However, yield stress of metals only expresses the bulk property of the tested samples. In sliding experiments, not only bulk properties but also surface properties of materials involved in the shear resistance. In his book "Friction and Wear of Materials", E. Rabinowicz has pointed out that surface energy plays more and more important role in tensile and hardness tests when the sample or indenter size is down to a few micrometers. For the nanoscale asperities investigated in the study, contributions of surface effect to shear resistance are not negligible.*

Response: We thank the reviewer for these insightful comments. We fully agree that surface energy and more broadly, the proximity of free surfaces or interfaces, can strongly affect mechanical properties at the nanoscale. In fact, when the surface-to-volume ratio of a region of interest is high, the yield stress is no longer a bulk property; as Rabinowicz pointed out and as has been subsequently discussed and investigated by many, the nucleation and propagation of dislocations (and thus, the yield strength) can be strongly affected by the proximity of free surfaces or interfaces. This is what we meant when we mentioned in the original manuscript,

on page 8, that “We observe effective shear stresses that reach 20-62% of this ideal value, despite having proximal surfaces and being measured at 300K which would both be expected to reduce the yield stress from the theoretical value” and then, “As well, the proximity of surfaces, and the temperature being finite, will lead to further reductions of the shear stress at yield from the ideal value of 1.65 GPa”. This issue is of key importance in the nanowire literature, including in the referenced studies of Ag nanowires. This was discussed, for example, in the originally submitted manuscript on page 9, where we stated that in reference 45, “...initial yield is correlated with particular plastic events, specifically, surface nucleation of stacking fault decahedrons.” In short, most theories of plasticity predict that free surfaces reduce strength; this makes our findings of high strength values all the more interesting. We have added a citation to recent work that comprehensively discusses these issues (Li, Q.-J.*et al.* (2018). *Sample-size-dependent surface dislocation nucleation in nanoscale crystals*. **Acta Mat.** 145, 19-29), reference 41 on page 8 of the revised manuscript, and updated the text along the lines the reviewer suggests to make this point clearer.

REVIEWERS' COMMENTS

Reviewer #1 (Remarks to the Author):

The authors have satisfactorily addressed my concerns. I recommend publication in Nature Communications.

Reviewer #2 (Remarks to the Author):

I am satisfied with the authors' replies.

Reviewer #3 (Remarks to the Author):

The revised manuscript has made all of my concerns clear. I recommend it to publish in Nature Communications with no reservation.